# Endocytic coelomocytes are required for lifespan extension by axenic dietary restriction

**Lucas Mergan◯, Brecht Driesschaert, Liesbet Temmerman***

Department of Biology, Animal Physiology and Neurobiology, University of Leuven (KU Leuven), Leuven, Belgium

* Liesbet.Temmerman@kuleuven.be

## Abstract

A rather peculiar but very potent means of achieving longevity is through axenic dietary restriction (ADR), where animals feed on (semi-)defined culture medium in absence of any other lifeform. The little knowledge we already have on ADR is mainly derived from studies using the model organism *Caenorhabditis elegans*, where ADR more than doubles organismal lifespan. What is underlying this extreme longevity so far remains enigmatic, as ADR seems distinct from other forms of DR and bypasses well-known longevity factors. We here focus first on CUP-4, a protein present in the coelomocytes, which are endocytic cells with a presumed immune function. Our results show that loss of *cup-4* or of the coelomocytes affects ADR-mediated longevity to a similar extent. As the coelomocytes have been suggested to have an immune function, we then investigated different central players of innate immune signalling, but could prove no causal links with axenic lifespan extension. We propose that future research focuses further on the role of the coelomocytes in endocytosis and recycling in the context of longevity.

## Introduction

Dietary restriction (DR), reducing food consumption without causing malnutrition, is a highly robust approach to increase lifespan and postpone age-related deterioration. It was first discovered in rats over 85 years ago [1], and has since been demonstrated in a wide variety of animals, ranging from *Caenorhabditis elegans* [2] to non-human primates [3]. In humans, DR decreases risk factors for diabetes, cardiovascular disease and cancer [4], and results in an improvement in general health, mood and sleep quality [5], amongst other benefits. However, its therapeutic use in humans is complicated by occasional unwanted side effects such as hypotension, reduced fertility and osteoporosis [6]. Therefore, unravelling molecular pathways upon which the advantageous effects of DR rely, may allow to exploit the health benefits while avoiding adverse side effects and the need for restrictive dietary lifestyle changes.

Thanks to the clear prolongevity effects of DR existing throughout the animal kingdom, model organism research has been able to advance our understanding already quite extensively [7,8]. Research in *C. elegans*, for example, warns against the umbrella concept of DR, because

633589, Horizon 2020, https://research-and-innovation.ec.europa.eu/ LT, G052217N, FWO Flanders LT, G049321N, FWO Flanders LT, C16/19/003, KU Leuven, https://www.kuleuven.be/ The funders had no role in study design, data collection and analysis, decision to publish, or preparation of the manuscript.

**Competing interests:** The authors have declared that no competing interests exist.

the numerous divergent methods to implement a DR regimen often rely on distinct mediating factors that do not always converge on one main pathway [9]. This shows that there is no simple answer as to how DR increases lifespan, and that studying different forms of DR diversifies the options to identify molecular players involved in ageing.

A specific form of DR that can be applied in *C. elegans* is axenic dietary restriction (ADR), in which worms are cultured in a sterile growth medium [10] and display slower development, reduced fecundity and a smaller body size [11]. Despite knowing for over two decades that ADR also leads to an impressive lifespan increase of at least two-fold [11], one can only say it seems not to require common ageing factors such as DAF-16a [12], SKN-1, PHA-4, HIF-1 or HSF-1 [13], and that it is not solely due to the absence of bacterial proliferation in the intestine, as raising worms on dead bacteria only results in a much smaller lifespan extension of 20–40% [14,15]. Therefore, the actual underlying mechanism remains enigmatic.

One of only two molecular players that has already been linked to ADR is the ligand-gated ion channel CUP-4, which when absent significantly reduces the longevity effect of ADR [13]. This protein was also found to be important in the longevity conferred by other forms of DR, specifically *eat-2* mutation and solid DR, in which it was suggested to be part of a pathway involving transcription factors PHA-4 and SKN-1 and neuropeptide NLP-7 [16]. However, this cannot be the case in ADR, as all three factors have been shown not to be involved in its lifespan extending effect [13]. CUP-4 is expressed in the *C. elegans* coelomocytes, where it plays a role in endocytosis of pseudocoelomic content [17]. The coelomocytes are six ovoid cells (five in males) of 10–15 μm in diameter, suspended in the pseudocoelomic cavity by attachment to the body wall muscle [18]. They are crucial to the lifespan extension seen in a variety of DR methods [19,20], but the underlying mechanism has not been determined, nor has their involvement in ADR been tested. Although their function has not been studied in as much detail as many other body systems of *C. elegans*, they have been suggested to serve as immune cells [21,22]. Recent work supports the involvement of immunity in longevity conferred by other forms of DR in *C. elegans* [23–26]. Here we find that the coelomocytes are important for the lifespan extension mediated by ADR, and we investigate the potential role of the three main innate immunity pathways in axenic longevity, but find no clear causal evidence for their involvement in ADR.

## Methods

### *C. elegans* strains

Table 1 lists all the strains used in this research. All in-house strains containing an extrachromosomal array were generated through microinjection [27]. LSC1987 was created by backcrossing N2 to KU25 six times, always selecting worms wild-type for *pmk-1* to cross with KU25. All strains were cultivated on 7.5 g/L peptone nematode growth medium (NGM) plates seeded with *Escherichia coli* OP50 at 20˚C [28].

### Liquid cultures

Fully fed (FF) medium consists of S complete medium supplemented with *E. coli* K12 (Artechno). S complete medium contains autoclaved S basal medium (10 mM NaCl, 5.7 mM $K_2HPO_4$ and 44 mM $KH_2PO_4$) supplemented with 3 mM $MgSO_4$, 3 mM $CaCl_2$, 5 μg/mL cholesterol, 10 mM potassium citrate, 55 μM disodium ethylenediaminetetraacetic acid (EDTA), 25 μM $FeSO_4$, 10 μM $MnCl_2$, 10 μM $ZnSO_4$, 1 μM $CuSO_4$, 50 units/mL nystatin (Sigma-Aldrich), 0.25 mg/mL penicillin, 0.25 mg/mL streptomycin and 0.5 mg/mL neomycin (PSN Antibiotic Mixture, Thermo Fisher Scientific). To feed the worms, previously snap-frozen *E. coli* K12 were added daily to an $OD_{600}$ of 1.87. Baffled Fernbach flasks containing at least 50

**Table 1. Strains used in this study.**

| Strain | Genotype | Source | Description |
|---|---|---|---|
| N2 | *wt* | *Caenorhabditis* Genetics Center (University of Minnesota, USA) | Bristol wild type |
| PHX1015 | *cup-4(syb1015)* | SunyBiotech (China) | CRISPR insertion of SL2::NLS:: GFP downstream of the *cup-4* ORF |
| LSC1961 | *cup-4(lst1684) III* | This work | CRISPR deletion of *cup-4* gene (944 bp promoter + ORF + 102 bp 3' UTR) |
| LSC1963 | *cup-4(lst1684) III; lstEx1065 [unc-122p::cup-4::unc-54utr 10 ng/µL; myo-2p::mCherry 1 ng/µL]* | This work | Coelomocyte-specific *cup-4* rescue (3978 bp *unc-122* promoter) in LSC1961 |
| LSC1964 | *cup-4(lst1684) III; lstEx1066 [cup-4p::cup-4::cup-4utr 10 ng/µL; myo-2p:: mCherry 1 ng/µL]* | This work | Endogenous *cup-4* rescue (1341 bp *cup-4* promoter) in LSC1961 |
| RB950 | *cup-4(ok837) III* | *Caenorhabditis* Genetics Center (University of Minnesota, USA) | 0x outcrossed *cup-4* knockout |
| NP717 | *unc-119 (ed3) III; arIs37[myo-3p::sel-1sp:: GFP]; cdIs32[unc-122p:: DT-A (E148D)::unc-54utr; unc-119(+); myo-2p::GFP]* | Hanna Fares (University of Arizona, USA) | Coelomocyte-ablated strain using diphtheria toxin under control of a 3978 bp *unc-122* promoter, and GFP secreted to the pseudocoelom using a 79 bp *sel-1* signal peptide |
| KU25 | *pmk-1(km25) IV* | *Caenorhabditis* Genetics Center (University of Minnesota, USA) | 6x outcrossed *pmk-1* knockout |
| NU3 | *dbl-1(nk3) V* | *Caenorhabditis* Genetics Center (University of Minnesota, USA) | 10x outcrossed *dbl-1* knockout |
| JV53 | *daf-16(jr22) I* | Bart Braeckman (Ghent University, Belgium) | CRISPR deletion of *daf-16* gene (bp 22.471–23.583, including four critical exons) |
| LSC1983 | *pmk-1(km25) IV; lstEx1076 [pmk-1p::pmk-1::pmk-1utr 20 ng/µL; myo-2p::mCherry 1 ng/µL]* | This work | Endogenous *pmk-1* rescue (2652 bp *pmk-1* promoter) in KU25 |
| LSC1984 | *pmk-1(km25) IV; lstEx1077 [unc-122p::pmk-1::unc-54utr 20 ng/µL; myo-2p::mCherry 1 ng/µL]* | This work | Coelomocyte-specific *pmk-1* rescue (3978 bp *unc-122* promoter) in KU25 |
| LSC1986 | *pmk-1(km25) IV; lstEx1079 [ges-1p::pmk-1::unc-54utr 10 ng/µL; myo-2p::mCherry 1 ng/µL]* | This work | Intestine-specific *pmk-1* rescue (2093 bp *ges-1* promoter) in KU25 |
| LSC1987 | *wt* | This work | N2 backcrossed to KU25 6x (wild-type *pmk-1* in KU25 background) |
| LSC2005 | *pmk-1(lst1688) IV* | This work | CRISPR deletion of *pmk-1* gene (447 bp promotor + ORF + 462 bp 3'-UTR) |

mL of medium with 1.0±0.2 worm/µL were kept at 20˚C in the dark and shaken at 121 rpm in an Innova 43 rotary shaker (New Brunswick).

Axenic medium consists of 3% Peptone N-Z Soy BL4 (Sigma) and 3% Bacteriological Grade Yeast Extract (VWR). After autoclaving, this is supplemented with 0.05% haemoglobin (diluted from a 5% stock solution in 0.1 M KOH) and 6.25 µg/mL cholesterol (diluted from a 10 mg/mL stock solution in ethanol). 50 units/mL nystatin (Sigma-Aldrich), 0.25 mg/mL penicillin, 0.25 mg/mL streptomycin, and 0.5 mg/mL neomycin (PSN Antibiotic Mixture, Thermo Fisher Scientific) are added to help prevent fungal and bacterial contamination. As this medium is highly nutritious for bacteria, it is crucial that all equipment is handled in a sterile manner in a laminar flow cabinet.

## Lifespan analysis

For all lifespan experiments, worms were synchronized. Gravid hermaphrodites were rinsed off their plates with S basal into a 15 mL tube and left to settle, after which the supernatant was

removed. 3 mL of S basal and 2 mL of a 1:2 solution of sodium hydroxide (5 M) and household bleach (18˚) were added, and the tube was vigorously shaken for 5 minutes. S basal was added to 15 mL and worms were pelleted at 350 rcf. The worms were washed a total of three times by removing supernatant, adding S basal and centrifuging. Eggs resuspended in 10 mL S basal were rotated at 20˚C overnight before transferring to plates seeded with OP50 *E. coli*. When an experiment included extrachromosomal transgenic strains, the synchronized population in L1 diapause was sorted using a complex object parametric analyser and sorter (COPAS, Union Biometrica) to select fluorescent worms. Non-fluorescent strains included in the same experiment were handled in a similar way and run through the machine using the same settings, but excluding the fluorescence gate. Following L1 diapause, worms were grown on NGM plates seeded with *E. coli* OP50 until the young adult stage, when worms were transferred to either FF or axenic conditions.

For solid FF, 15 worms were picked onto 55 mm NGM plates seeded with 50 μL *E. coli* OP50 (grown overnight at 37˚C in LB medium) and supplemented with 200 μM 5-fluorodeoxyuridine (FUdR); ten plates were prepared per condition. For liquid FF, worms were rinsed off their plates using S basal and left to settle, after which the worm pellet was pipetted onto unseeded NGM plates; five worms were then picked per well containing 200 μL S complete medium supplemented with 100 μM FUdR and *E. coli* HB101 at an $OD_{600}$ of 1.95, and this for 30 wells per condition. For ADR, worms were rinsed off their plates using S basal, and the worm pellet was pipetted onto unseeded plates; five worms were picked per tube containing 300 μL axenic medium supplemented with 100–200 μM FUdR, and this for 25 tubes per condition.

For solid and liquid FF, live worms were counted daily by prodding immobile worms with a picker to determine movement. For ADR, survival was evaluated every two days by tapping tubes to determine movement. Survival curves were analysed using Weibull accelerated failure time due to non-constant hazards [29], corrected for multiple testing using the Benjamini-Hochberg method [30].

The relative importance of a gene (knockout) or cell (ablation) in ADR was calculated using the following formula on the mean lifespans of control (c) and test (t) populations:

$$Relative\ importance = \left( \frac{ADR_c}{FF_c} - 1 \right) - \left( \frac{ADR_t}{FF_t} - 1 \right)$$

## Fluorescence microscopy

Worms were synchronized as described under *Lifespan analysis*. Upon reaching early adulthood (±48 h post L1 diapause), worms were rinsed off their plates, washed three times with S basal and transferred to 50 mL of either FF or ADR medium containing 100 μM FUdR, 0.5 mg/mL penicillin, 0.5 mg/mL streptomycin, and 1 mg/mL neomycin (PSN Antibiotic Mixture, Thermo Fisher Scientific). Day 2 adult worms (±96 h post L1 diapause) were mounted on 2% agarose pads and anesthetized with 10 μL of 1 mM sodium azide. Worms were imaged using a DM6 B microscope (Leica) with GFP filters.

## Results

### *cup-4* and the coelomocytes are required for ADR longevity

To reveal more about the possible mechanisms of ADR-induced longevity, we focused on *cup-4*, which has already been linked to ADR [13]. However, available evidence relies on a non-outcrossed partial *cup-4* deletion mutant *(ok837)* that was generated using random mutagenesis, leaving the possibility that a background mutation might be causally linked to the observed

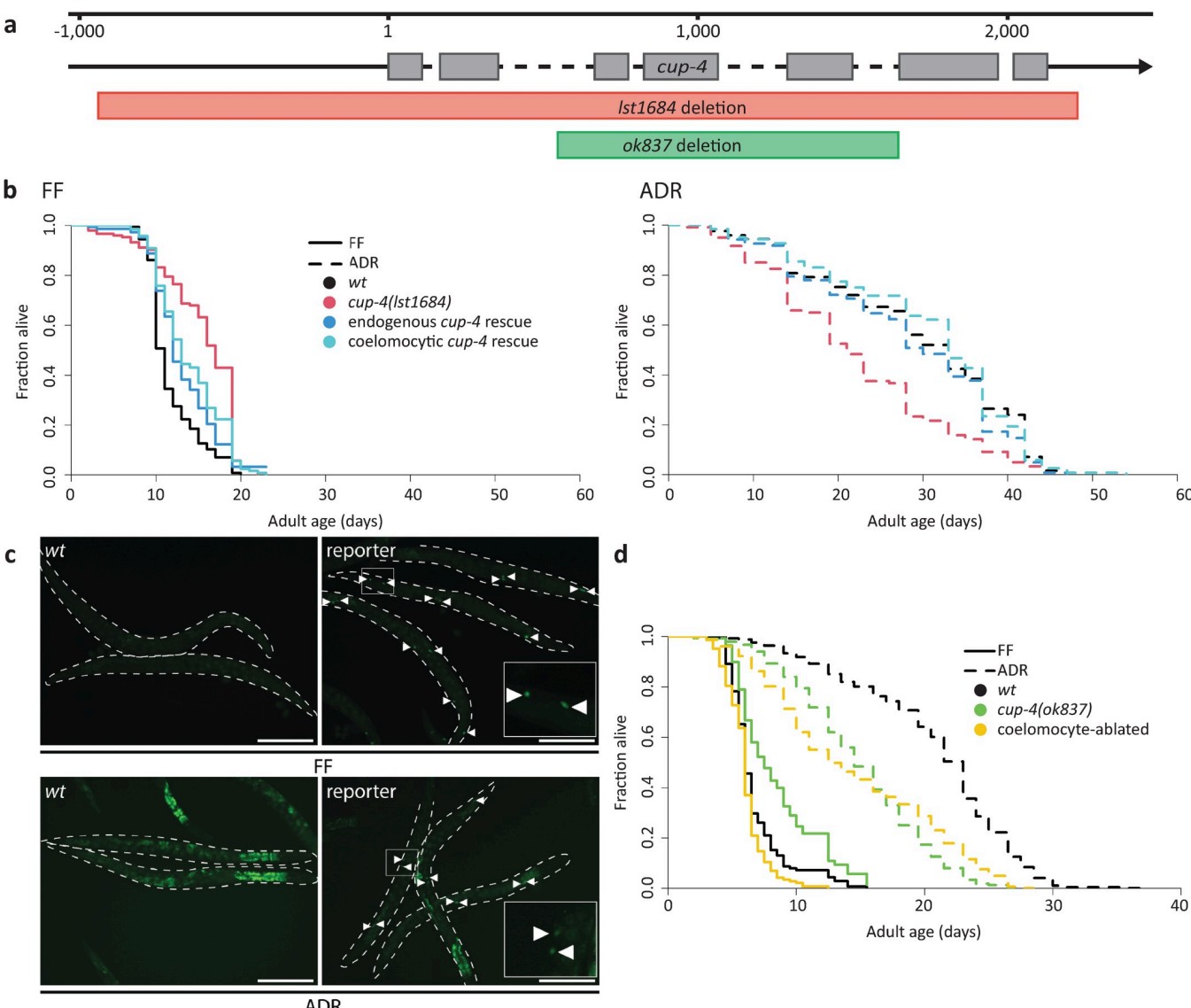

**Fig 1. *cup-4* is required specifically in the coelomocytes for axenic lifespan extension. A.** Gene map of *cup-4* indicating the deleted regions of alleles used in this study. **B.** Lifespan in solid fully fed (FF, left) and liquid axenic dietary restriction (ADR, right) conditions of wild-type, *cup-4(lst1684)* deletion, endogenous *cup-4* rescue (*cup-4p*) and coelomocyte-specific *cup-4* rescue (*unc-122p*) worms. In FF, *cup-4(lst1684)* (p = 2.0E-15) is long-lived and both endogenous (p = 1.9E-04) and coelomocyte-specific (p = 4.5E-08) rescues are slightly long-lived compared to wild type. In ADR, *cup-4(lst1684)* is strongly short-lived compared to wild type (p = 5.2E-06) while both endogenous (p = 0.47) and coelomocyte-specific (p = 0.60) rescues do not significantly differ from wild type. **C.** Bright green fluorescence can be seen in the coelomocytes of a *cup-4(syb1015)* reporter worm in both FF and ADR (white arrows), alongside wild-type levels of green autofluorescence in the gut. Green autofluorescence is seen in the gut of a wild-type worm in both FF and ADR, although more strongly so in ADR. Scale bar is 250 μm. **D.** Lifespan in liquid FF and liquid ADR conditions of wild-type, *cup-4(ok837)* deletion and coelomocyte-ablated worms. In FF, *cup-4(ok837)* is long-lived (p = 1.6E-10) while the coelomocyte-ablated strain is slightly short-lived (p = 7.1E-06) compared to wild type. In ADR, both *cup-4(ok837)* (p = 8.0E-16) and coelomocyte-ablated worms (p = 4.0E-16) are short-lived compared to wild type. Mean lifespans, sample numbers and statistical significance are shown in S1 Table.

phenotype. Hence, we used CRISPR-Cas9 in our laboratory stock of the CGC N2 background to create a mutant in which the promoter and open reading frame of *cup-4* were deleted (Figs 1A and S1). This *cup-4(lst1684)* knockout mutant lived somewhat longer than wild type when fully fed, but much shorter than wild type in ADR conditions (Fig 1B), similar to what has been reported for *cup-4(ok837)* [13], and resulting in a high relative importance of 1.08 (S1 Table). This means that *cup-4* has a huge effect, since for these mutants the observed ADR

lifespan extension is 108 percent points shorter than for wild types, *i.e.* an extension in ADR of only 44% instead of 152%. We could also restore ADR longevity to control levels when extra-chromosomally rescuing *cup-4,* confirming loss of this gene is indeed causal to the observed effect (Figs 1B and S1).

Because *cup-4* expression is enriched in coelomocytes [17], we next asked whether it is the coelomocytic *cup-4* that is essential to support ADR-mediated longevity. Via targeted genetics, we first introduced SL2::NLS::GFP directly after the *cup-4* open reading frame, creating a strain that produces GFP reporters alongside CUP-4 at endogenous levels. Green fluorescence can be seen brightly in all six coelomocytes under fully fed or dietary restricted conditions alike, confirming the localization of *cup-4* to these cells (Fig 1C). While we cannot rule out low levels of *cup-4* expression in other cells, we did not observe reporter expression anywhere else. In addition, reintroducing *cup-4* specifically in the coelomocytes of *cup-4* deletion mutants, could fully restore ADR longevity (Figs 1B and S1), confirming the importance of coelomocytic *cup-4* in ADR.

Given the results described above, it would be likely for the coelomocytes themselves to be essential to ADR-induced longevity. Indeed, animals in which the coelomocytes are ablated, display an ADR lifespan similar to those of *cup-4* deletion mutants (Fig 1D). Under FF conditions, coelomocyte ablation results in a slight lifespan decrease, while *cup-4(ok837)* mutants have a longer lifespan than wild type. Both strains have a much shorter lifespan than wild type under ADR. This results in high relative importances of 1.32 for *cup-4(ok837)* and 0.68 for coelomocyte-ablated animals, confirming that the coelomocytes are required for axenic lifespan extension (Fig 1D, S1 Table). It appears from these data that for axenic conditions, losing *cup-4* versus losing the cells it is expressed in, results in a similar lifespan outcome. However, losing *cup-4* has a positive effect on FF lifespan that is abrogated when the coelomocytes are ablated, explaining the higher relative importance for *cup-4* mutation *vs* coelomocyte ablation.

## Insulin signalling and p38-MAPK are not required for ADR longevity

Because the coelomocytes are proposed immune cells [21,22], we wondered whether immunity pathways may affect ADR longevity. The *C. elegans* innate immune response is mediated primarily through three major pathways [31], homologous to the insulin/insulin-like growth factor signalling (IIS) cascade [32], the transforming growth factor beta (TGF-β) cascade [33], and the mammalian p38 mitogen-activated protein kinase (p38-MAPK) cascade [34]. To probe for a possible contribution of these to ADR longevity, lifespans were performed in FF and ADR upon knockout of central players for each of these pathways: *daf-16*, *dbl-1*, or *pmk-1*, respectively.

*daf-16(jr22)* is 25% shorter lived compared to wild type in FF conditions and maintains this difference under ADR (26% shorter lived than wild type), resulting in a negligible relative importance of 0.05 towards ADR-mediated longevity (Fig 2A, S1 Table).

In our hands, *dbl-1(nk3)* displays a 13% longer FF lifespan and under ADR becomes slightly shorter lived than wild type, now displaying a small but significant reduction in lifespan of 6% (Fig 2A). However, our FF result opposes observations by others [33,35–38], and previous work points towards an interaction effect between FUdR–used in all our experiments to avoid progeny–and *dbl-1* mutation [33,38]. We confirmed this interaction effect for FF conditions, with a lower concentration of FUdR resulting in a shorter lifespan (S2 Fig). Hence, these results require additional research to resolve what, if any, the specific relative importance of the *dbl-1* branch of immunity might be for ADR longevity.

The KU25 *pmk-1(km25)* strain displays a wild-type FF lifespan, but lives 20% shorter than wild type under ADR, resulting in a relative importance of 0.44 (Fig 2A, S1 Table). We

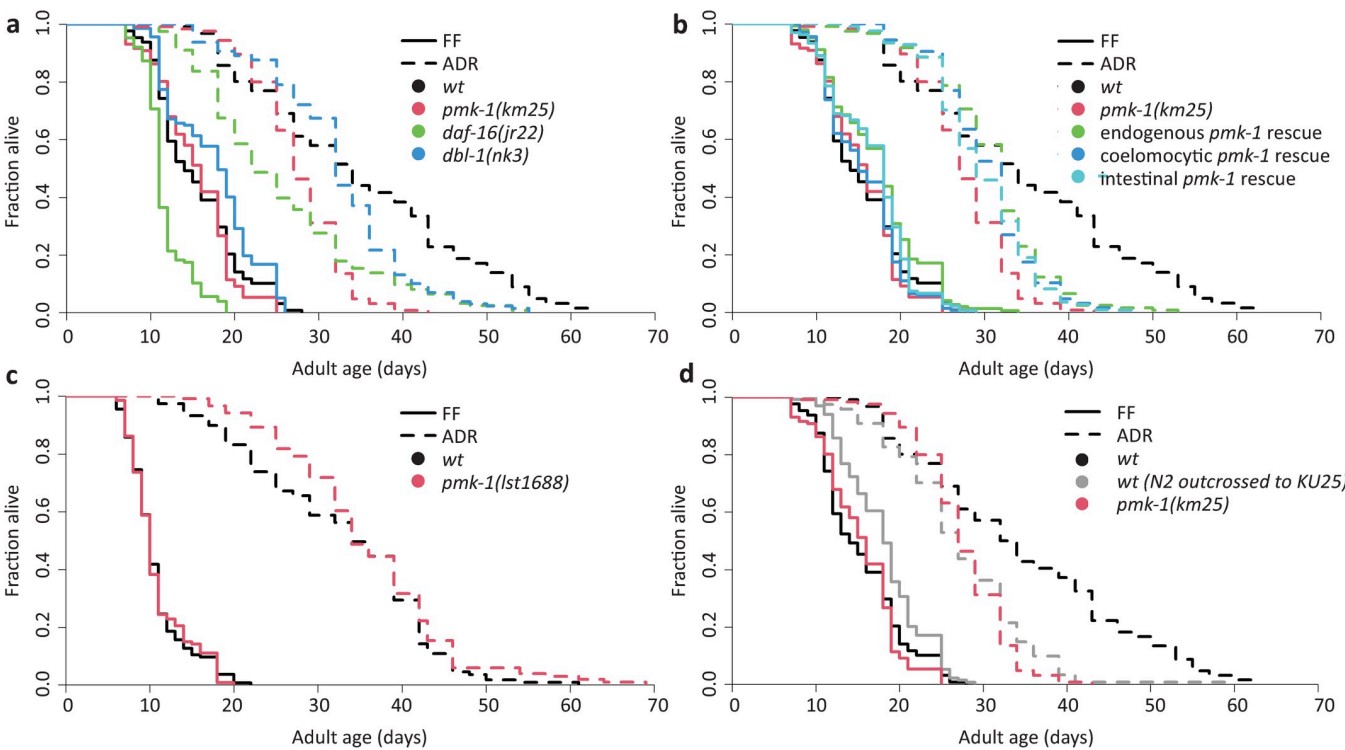

**Fig 2.** *daf-16* **and** *pmk-1* **are dispensable to axenic lifespan extension. A.** Lifespan in liquid fully fed (FF) and liquid axenic dietary restriction (ADR) conditions of wild-type and of *pmk-1(km25)*, *daf-16(jr22)*, and *dbl-1(nk3)* deletion worms. *pmk-1(km25)* does not significantly differ from wild type under FF (p = 0.34), but is short-lived under ADR (p = 1.8E-15). *daf-16(jr22)* is similarly short-lived under FF (p = 3.6E-15) and ADR (p = 1.2E-15). *dbl-1(nk3)* is slightly long-lived under FF (p = 0.0059) and slightly short-lived (p = 1.9E-05) under ADR, but an interaction affect with FUdR is likely (S2 Fig). **B.** Lifespan in liquid FF and liquid ADR conditions of wild-type, *pmk-1(km25)* deletion, endogenous *pmk-1* rescue (*pmk-1p*), coelomocyte-specific *pmk-1* rescue (*unc-122p*) and intestine-specific *pmk-1* rescue (*ges-1p*) worms. In ADR, *pmk-1(km25)* is short-lived (p = 1.8E-15) compared to wild type, as are the endogenous (p = 1.7E-10), coelomocyte-specific (p = 1.6E-13) and intestine-specific (p = 9.0E-16) rescues. **C.** Lifespan in liquid FF and liquid ADR conditions of wild-type and *pmk-1 (lst1688)* deletion worms. No significant difference was observed under FF (p = 0.95) or ADR (p = 0.19). Mean lifespans, sample numbers and statistical significance are shown in S1 Table. **D.** Lifespan in liquid FF and liquid ADR conditions of wild-type, wild-type *pmk-1* in KU25-enriched background, and *pmk-1(km25)* deletion worms. The worms with a functional *pmk-1* in a KU25 background have a similar axenic lifespan to KU25 *pmk-1(km25)* knockout worms (p = 0.23), while both have a significantly shorter axenic lifespan than wild-type worms (p = 1.2E-15/p = 6.0E-16).

therefore tested whether PMK-1 might be more active under ADR; this appears not to be the case at a whole-worm level (S3 Fig, S1 Methods). With *pmk-1* nevertheless emerging as the most promising candidate, we asked if it would be needed in the coelomocytes for ADR lifespan extension. In literature, *pmk-1* expression has been reported in the intestine and in head neurons [39]. Its immune function seems to be linked mainly to its intestinal expression [40–42], therefore, besides the coelomocytes, the intestine is a likely site of action for PMK-1 in the longevity conferred by ADR. We therefore measured FF and ADR lifespans of both coelomocyte- and intestine-specific *pmk-1* rescues alongside an endogenous rescue (Fig 2D, S1 Table, S1 Fig). Against what would be expected in case of causality (S1 Fig), endogenously rescuing *pmk-1* only had a small effect on ADR lifespan, with wild-type lifespan not being regained. Moreover, a similar small extension was seen under FF conditions, suggesting this is independent of the dietary condition. Similar effects were seen for coelomocyte- and intestine-specific rescues (Fig 2D), although statistical significance was below cut-off in the case of the coelomocyte-specific rescue in FF conditions (S1 Table). These results suggest that the axenic lifespan effect seen in *pmk-1(km25)* may rather result from a background mutation, or perhaps the expression of a truncated PMK-1 protein in this partial deletion mutant. To test this, a new *pmk-1* full gene deletion mutant was created in our laboratory N2 stock using CRISR-Cas9

technology. No lifespan defect was observed in both FF and ADR for this *pmk-1(lst1688)* mutant, resulting in a negative and small relative importance of -0.25 (Fig 2E, S1 Table). This suggests that the lifespan effect observed in the KU25 *pmk-1(km25)* strain is not due to the absence of PMK-1 protein, and that *pmk-1* is not involved in axenic longevity (S1 Fig). This was further confirmed in a strain created by backcrossing an N2 wild type *pmk-1* allele into the KU25 background (six times backcrossed). This strain had a shortened axenic lifespan similar to *pmk-1(km25)* despite it having an intact *pmk-1* gene (Fig 2F), again showing that *pmk-1* does not causally contribute to ADR longevity (S1 Fig).

## Discussion

Despite it being the most potent form of dietary restriction identified in *C. elegans* to date, the mechanisms underlying axenic longevity remain undiscovered, with many known factors (DAF-16a, SKN-1, PHA-4, HIF-1 or HSF-1) already ruled out as possible causative mechanisms [12,13]. The only two factors suggested to play a role, are the transcriptional coactivator CBP-1 [43,44] and the ligand-gated ion channel CUP-4 [13]. The former has already been studied in some detail [44], while until this work, the latter had not been investigated further regarding its role in ADR. We here confirmed the importance of CUP-4 and localized its site of action to the coelomocytes, which themselves we also show to be essential to ADR longevity.

It has been hypothesized that coelomocytes perform an immune function in *C. elegans* [21,22], which is supported by the immune role of their homologues in other invertebrates, including closely related nematodes [45–48]. On a genetic level, some parallels can also be drawn between *C. elegans* coelomocytes and mammalian macrophages [49–52]. Considering the potential immune function of the coelomocytes, we asked whether immunity might be playing a role in axenic lifespan extension. This would not be unreasonable, since whole-mount transcriptomic data show an upregulation of immune genes under ADR [53]. Immunity is known to be involved in other forms of DR [23–26], and immunity has been linked to ageing in *C. elegans*, as older animals have a decreased resistance to bacterial infections [54] and p38-MAPK immune signalling is impaired during ageing [55]. However, we could not find convincing evidence for any of the major immune pathways to contribute to axenic longevity in *C. elegans*. For the IIS pathway, it is already known that the *daf-16a* isoform is dispensable to ADR [12], and that *daf-2* longevity is additive to that of ADR [11]. Our *daf-16 (jr22)* knockout of all isoforms had a shortened lifespan similar to reported lifespans of other *daf-16* alleles [56–58], and we observed a shortening under ADR to a similar extent, ruling out a role for IIS in ADR. For TGF-β signalling, unfortunately, due to interaction effects with FUdR, no solid conclusions can yet be drawn. While it would be useful to remove FUdR from the equation, the practicalities of liquid FF and ADR lifespan assays complicate this beyond what is experimentally feasible, leaving details on this immune branch unresolved at the moment. Finally, the p38-MAPK pathway was investigated using *pmk-1* knockouts. A wild-type FF lifespan was observed, in accordance with the majority of literature [37,59–61], but no causal longevity effect was present in ADR either. However, we showed that the KU25 strain seems to carry one or multiple background mutations that do influence ADR longevity, which from a forward genetics perspective may permit to provide interesting leads for future research.

Based on our findings, we propose that the coelomocytes play an important role in axenic longevity that does not depend on immune pathways. Considering that the main known activity of these cells is the endocytosis and breakdown of material from the pseudocoelom, and blocking this endocytosis through knockout of *cup-4* abrogates ADR longevity to a similar extent as coelomocyte ablation, we suggest that it is their role in recycling that may be essential

to lifespan extension. This would fit well within the garbage accumulation theory, which argues that the ability of the body to clear waste products diminishes during ageing, leading to a build-up of indigestible detritus which hinders correct cellular functioning [62]. There is evidence that ageing worms accumulate yolk proteins [63], fluorescent pigments [2], and aggregates of insoluble proteins homologous to human disease aggregates [64] in their body cavity, which may contribute to unhealthy ageing [65,66], although causality is debated [67]. This build-up of detritus is partially kept in check by extracellular proteostasis [68], but it may also be controlled by the coelomocytes, with their role becoming even more important in long-lived worms, as they have a lengthier lifetime over which to accumulate harmful waste. It would be interesting to see the effect of overexpression of *cup-4* in both FF and ADR conditions, to determine whether heightened endocytosis increases lifespan in either of these conditions. Alternatively, the coelomocytes may be required to provide recycled nutrients; in ADR, insufficient nutrients may be taken up from the environment, meaning recycling of internal material is needed to provide building blocks, while a fully fed worm can simply get these from its food source. A next step to unveil whether endocytosis is the reason that CUP-4 and the coelomocytes are crucial to ADR, could involve testing the axenic lifespan of other mutants in which coelomocytic endocytosis is defective. The coelomocytes have also been implicated in other lifespan-extending dietary interventions [19,20], as has the CUP-4 protein [16], and it would be interesting to unveil whether they might be equally important in non-dietary (*e.g.* genetic or pharmaceutical) interventions, perhaps uncovering a more general relevance to lifespan extension due to their role in keeping the body fluids in a clean state and/or providing recycled building blocks. If the coelomocytes can be established as important regulators of waste accumulation and/or recycling during ageing, they could provide a strong model to investigate these fundamental principles in a simple anatomical context.

## Supporting information

**S1 Fig. Experimental rationale for *cup-4* and *pmk-1* experiments.** When a phenotype is observed in a genetic knockout mutant, different approaches can be used to further prove causality, as shown here. Outcomes in bold are those that are conclusive, others form strong indications. In our experiments, an ADR lifespan phenotype was initially observed for *cup-4 (ok837)* and *pmk-1(km25)*. Further steps taken and their outcome are marked with C (*cup-4*) and P (*pmk-1*).
(TIF)

**S2 Fig. FUdR concentration influences the lifespan of a *dbl-1* mutant.** Lifespan in FF conditions of *dbl-1(nk3)* worms at a concentration of either 50 μM or 100 μM FUdR. At a lower FUdR concentration, *dbl-1(nk3)* mutants are significantly shorter lived (p = 1.1E-04) than at the higher concentration. Mean lifespans, sample numbers and statistical significance are shown in S1 Table.
(TIF)

**S3 Fig. phospho-PMK-1 levels are not significantly changed under ADR. A.** Uncropped western blot and Coomassie total protein stain of three biological replicates under fully fed (FF) and axenic dietary restriction (ADR) conditions. phospho-PMK-1 (p-PMK-1) band is marked by an arrow. **B.** Quantification of p-PMK-1 levels as measured by Western blot and normalized to total protein. PMK-1 activity was not increased under ADR, as no significant difference was observed based on an unpaired two-samples *t*-test, with a possible trend to lower activity levels under ADR.
(TIF)

**S1 Table. Data and statistics from all lifespan experiments.** As hazards were not constant, Weibull accelerated failure time was used for P-value calculation. Benjamini-Hochberg correction was used for multiple comparisons. ns not significant * p<0.05 ** p<0.01 *** p<0.001. (XLSX)

**S1 Methods. p38-MAPK activity assay.**
(PDF)

**S1 Raw images. Gel images used in S3 Fig.**
(TIF)

## Acknowledgments

We thank Bart Braeckman and Lieselot Vandemeulebroucke for guidance regarding axenic culturing and Francisco Naranjo Galindo, Luc Vanden Bosch, Elke Vandewyer, Amanda Kieswetter and Marijke Christiaens for technical laboratory assistance.

## Author Contributions

**Conceptualization:** Lucas Mergan, Brecht Driesschaert, Liesbet Temmerman.

**Formal analysis:** Lucas Mergan, Brecht Driesschaert.

**Funding acquisition:** Liesbet Temmerman.

**Investigation:** Lucas Mergan, Brecht Driesschaert, Liesbet Temmerman.

**Methodology:** Lucas Mergan, Brecht Driesschaert, Liesbet Temmerman.

**Resources:** Liesbet Temmerman.

**Supervision:** Liesbet Temmerman.

**Visualization:** Lucas Mergan.

**Writing – original draft:** Lucas Mergan.

**Writing – review & editing:** Lucas Mergan, Brecht Driesschaert, Liesbet Temmerman.

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
