## [Decision Letter · Decision Letter 0]

25 May 2023

PONE-D-23-10548Endocytic coelomocytes are required for lifespan extension by axenic dietary restrictionPLOS ONE

Dear Dr. Mergan,

Thank you for submitting your manuscript to PLOS ONE. After careful consideration, we feel that it has merit but does not fully meet PLOS ONE’s publication criteria as it currently stands. Therefore, we invite you to submit a revised version of the manuscript that addresses the points raised during the review process.

We look forward to receiving your revised manuscript.

Kind regards,

Adler R. Dillman, Ph.D.

Academic Editor

PLOS ONE

When you resubmit, please ensure that you provide the correct grant numbers for the awards you received for your study in the ‘Funding Information’ section."

Additional Editor Comments:

After careful reading of the manuscript and the reviewer comments, it is clear that major revisions are required. Reviewer 1 points out that more background on the dietary restriction field would be helpful and the significance of highly relevant previous literature that was not cited nor considered in the current manuscript. The authors should read Park et al., 2010 and other papers from that group that are relevant to dietary restriction, axenic dietary restriction, and cup-4. The difference between the previous work and this current manuscript should be discussed. Originality of work is an important criterion for PLOS One; "If a submitted study replicates or is very similar to previous work, authors must provide a sound scientific rationale for the submitted work and clearly reference and discuss the existing literature. Submissions that replicate or are derivative of existing work will likely be rejected if authors do not provide adequate justification."

Reviewer 2 suggests several experiments that could be done to address how much of the reduction in ADR-mediated lifespan extension observed upon coelomocyte ablation is accounted for by CUP-4. Previous work may help address this question, and while I agree that these suggestions by Reviewer 2 are interesting and important, they are not necessary in revising the current manuscript.

Reviewers' comments:

Reviewer's Responses to Questions

**Comments to the Author**

1. Is the manuscript technically sound, and do the data support the conclusions?

Reviewer #1: Partly

Reviewer #2: Yes

2. Has the statistical analysis been performed appropriately and rigorously? 

Reviewer #1: I Don't Know

Reviewer #2: Yes

3. Have the authors made all data underlying the findings in their manuscript fully available?

Reviewer #1: No

Reviewer #2: Yes

4. Is the manuscript presented in an intelligible fashion and written in standard English?

Reviewer #1: Yes

Reviewer #2: Yes

5. Review Comments to the Author

Reviewer #1: It is widely known that both dietary restriction (DR) and axenic culture can increase the lifespan of C. elegans. Axenic dietary restriction (ADR) would probably have the combined effect of both conditions. Therefore, it should be reasonable to discuss what is known in DR studies that are highly related to this research.

A previous paper (Park et al., 2010) (and other papers from their group) suggest that cup-4 and coelomocytes are required for dietary restriction (DR) associated life extension.

Park, S. K., C. D. Link and T. E. Johnson, 2010 Lifespan extension by dietary restriction is mediated by NLP-7 signaling and coelomocyte endocytosis in C. elegans. FASEB J 24: 383-392.

The paper should have been cited, and the difference between this manuscript and the paper should have been discussed.

When discussing other genes in the paper, the authors also failed to discuss or mention highly relevant works that had been done in the DR field.

It is also debatable whether the effect observed by the authors between fully fed (bacteria-fed) and ADR were caused by axenic, DR, a combination effect, or just simply different food source and nutrient content. A more fair comparison would be perhaps using killed bacteria.

In Fig. 1B, since the fully fed is grown on solid mediums and the ADR in liquid, it is probably better to use separate plots to present the data as the solid and liquid medium is known to have a significant effect on the worm’s development.

Reviewer #2: In this manuscript Mergan et al. demonstrate 1. that CUP-4 acts in coelomocytes to regulate lifespan extension under axenic dietary restriction and 2. that the immune regulator PMK-1 is dispensable for ADR-mediated lifespan extension. The construction and use of CRISPR mutant alleles for cup-4 and pmk-1 strengthens the conclusions about the involvement of cup-4 and the dispensability of pmk-1 in ADR-mediated longevity. It is important to address how much of the reduction in ADR-mediated lifespan extension observed upon coelomocyte ablation is accounted for by CUP-4, and whether other immune pathways act independently of coelomocytes to regulate lifespan upon ADR. I make some suggestions below for addressing these:

1. Does endogenous CUP-4 rescue completely rescue the reduction in ADR-mediated lifespan extension in coelomocyte-ablated animals?

2. Do other components involved in coelomocyte endocytosis regulate ADR-mediated lifespan extension?

3. (a) The dbl-1 mutant shows a reduction in ADR-mediated lifespan extension – does this effect persist in a lifespan assay performed without FuDR?

(b) Are the effects of dbl-1 and coelomocyte ablation (or cup-4 loss-of-function) on the extended lifespan under ADR additive, i.e., do DBL-1 and coelomocyte endocytosis act independently to regulate lifespan under ADR? If so, then the paper can be about how coelomocyte endocytosis and innate immune pathways independently regulate lifespan upon ADR.

4. How does CUP-4 regulate lifespan extension in ADR? Does LIPL-5 in coelomocytes or intestinal fat content change upon ADR and do these involve CUP-4 or coelomocyte endocytosis?

5. (minor) Does cup-4 expression in wt background extend wt lifespan?

I also suggest the following text and figure changes for clarity:

1. The authors have included p-values in Figure legends – they can consider also including them in the figures on top of the bar plots or next to the survival curves, at least for some of the comparisons.

2. Page 2, line 36 – consider replacing ‘DAF-16a’ with ‘DAF-16’

3. In Figure 2d, consider replacing ‘wt (backcrossed)’ with’ wt (KU25 outcrossed to N2)’

4. Figure S1 – typo: replace ‘oberved’ with ‘observed’

5. Page 10, lines 166-167; if the authors do not find a link between coelomocyte endocytosis and other immune pathways and instead discover that they act independently (see experiments suggested above) they can consider rephrasing these sentences to ‘Coelomocytes are immune cells which ‘phagocytose’ pseudocoelomic content. We wondered whether other immune pathways are involved in ADR-mediated longevity.

6. Page 9, line 151 – ‘localization’ instead of ‘localisation’

7. Page 10, line 183 – typo: change to ’slightly short-lived (p=1.9E-05) under ADR, but an interaction effect with FUdR is likely’

8. Page 13, line 241 – ‘hypothesized’ instead of ‘hypothesised’

9. Figure 1 – consider rearrange panels? The order of C and D are confusing.

10. For Figure S2, consider referring to Table S1.

6. PLOS authors have the option to publish the peer review history of their article (what does this mean?). If published, this will include your full peer review and any attached files.

Reviewer #1: No

Reviewer #2: No

---

## [Author Response · Author response to Decision Letter 0]

2 Jun 2023

We have provided a detailed response in the "Response to Reviewers" file attached to our submission.

---

## [Editor Report · Decision Letter 1]

15 Jun 2023

Endocytic coelomocytes are required for lifespan extension by axenic dietary restriction

PONE-D-23-10548R1

Dear Dr. Mergan,

We’re pleased to inform you that your manuscript has been judged scientifically suitable for publication and will be formally accepted for publication once it meets all outstanding technical requirements.

Kind regards,

Adler R. Dillman, Ph.D.

Academic Editor

PLOS ONE

Additional Editor Comments (optional):

Thank you for submitting this revised manuscript, and for addressing point-by-point the reviewers' concerns. This is fine work and helps move the field forward.
---

## [Editor Report · Acceptance letter]

19 Jun 2023

PONE-D-23-10548R1 

Endocytic coelomocytes are required for lifespan extension by axenic dietary restriction 

Dear Dr. Mergan:

I'm pleased to inform you that your manuscript has been deemed suitable for publication in PLOS ONE. Congratulations! Your manuscript is now with our production department. 

Kind regards, 

on behalf of

Dr. Adler R. Dillman 

Academic Editor

PLOS ONE